# Overview of First-Line and Second-Line Pharmacotherapies for Osteoarthritis with Special Focus on Intra-Articular Treatment

**DOI:** 10.3390/ijms23031566

**Published:** 2022-01-29

**Authors:** Alicja Nowaczyk, Dawid Szwedowski, Ignacio Dallo, Jacek Nowaczyk

**Affiliations:** 1Department of Organic Chemistry, Faculty of Pharmacy, LudwikRydygier Collegium Medicum in Bydgoszcz, Nicolaus Copernicus University in Toruń, 2 dr. A. Jurasza St., 85-094 Bydgoszcz, Poland; 2Department of Orthopaedics and Trauma Surgery, Provincial Polyclinical Hospital, 87-100 Toruń, Poland; dszwedow@yahoo.com; 3Orthopedic Arthroscopic Surgery International (O.A.S.I.) Bioresearch Foundation, Gobbi N.P.O., 20133 Milan, Italy; 4Unit of Biological Therapies, SportMe Medical Center, Department of Orthopaedic Surgery and Sports Medicine, 41013 Seville, Spain; doctorignaciodallo@gmail.com; 5Department of Physical Chemistry and Physicochemistry of Polymers, Faculty of Chemistry, Nicolaus Copernicus University, 7 Gagarina St., 87-100 Toruń, Poland

**Keywords:** osteoarthritis (OA), pharmacotherapies, local treatment, disease-modifying drugs for osteoarthritis (DMOADs)

## Abstract

Osteoarthritis (OA) can be defined as the result of pathological processes of various etiologies leading to damage to the articular structures. Although the mechanism of degenerative changes has become better understood due to the plethora of biochemical and genetic studies, the drug that could stop the degenerative cascade is still unknown. All available forms of OA therapy are based on symptomatic treatment. According to actual guidelines, comprehensive treatment of OA should always include a combination of various therapeutic options aimed at common goals, which are pain relief in the first place, and then the improvement of function. Local treatment has become more common practice, which takes place between rehabilitation and pharmacological treatment in the hierarchy of procedures. Only in the case of no improvement and the presence of advanced lesions visible in imaging tests, should surgery be considered. Currently, an increasing number of studies are being published suggesting that intra-articular injections may be as effective or even more effective than non-steroidal anti-inflammatory drugs (NSAIDs) and result in fewer systemic adverse events. The most commonly used preparations are hyaluronic acid (HA), glucocorticosteroids (GS), and also platelet-rich plasma (PRP) in recent years. This review aims to present the mechanism of action and clinical effectiveness of different pharmacological options in relieving pain and improving functions in OA as well as the emerging approach in intra-articular treatment with PRP.

## 1. Introduction

Arthritis is classified into various types, including osteoarthritis, rheumatoid arthritis (RA, ICD-10-CM code: M06 [1]), gouty arthritis, septic arthritis (SA), and psoriatic arthritis (PsA). Among them, RA and OA are the most common types of arthritis in the human population affecting about 302 million people [2]. Hallmarks of RA include inflammatory changes in the synovial membranes and articular structures, widespread fibrinoid degeneration of the collagen fibers in mesenchymal tissues as well as atrophy and rarefaction of bony structures. RA can also affect body parts other than joints, such as the eyes, mouth, and lungs. The etiology of RA is unknown, but most commonly autoimmune mechanisms have been implicated. In autoimmune diseases, the body’s immune cells attack the body’s tissues [3]. On the other hand, OA (ICD-10-CM code: M15-M19 [4]) is a complex, multifactorial disease. It can occur in any joint, mainly targeting the major joints (knee, hip, and back); nevertheless, it commonly affects hands, elbows, and ankles [5]. It is characterized by degeneration of the articular cartilage that ultimately leads to joint destruction [6]. The pathological changes of OA involve the entire joint, which is composed of multiple types of tissues and cells, exemplified by cartilage degradation, subchondral bone thickening, osteophyte formation, synovium inflammation, and hypertrophy, as well as ligament degeneration [7,8]. Both RA and OA commonly cause chronic pain, reduce motion in joints, and will become more prevalent given the aging population [9]. They have several underlying causes, including biochemical and mechanical factors [10,11]. In general, these diseases are progressive and no widely applied effective therapy is available [12].

So far, arthritis is managed pharmacologically and non-pharmacologically, including common pharmacotherapies, surgery, and lifestyle changes. Initially, the therapy should be based on non-pharmacological methods, such as weight reduction, improvement of special exercises, and orthoses. This first step is often overlooked or insufficiently emphasized, to the detriment of patients [13]. Pain medications, including the most popular NSAIDs, are the first-line treatment [14]. However local treatment has been used more widely, which in the hierarchy of proceedings takes place between rehabilitation and pharmacological treatment. Surgery should be considered only in the case of insufficient improvement and the presence of advanced lesions visible in imaging tests.

In this paper, we present an overview of the first- and second-line pharmacotherapy of OA. Due to the main goal, we discuss the literature reports regarding issues related to drug-based therapies as well as platelet-rich plasma. The work aims to present the reader with a wide range of systematized information concerning arthritis treatment. To enable a better understanding of the discussed issues, chemical structures of selected drugs are shown in the figures. Additionally, growth factors (GF) and cytokines of PRP are gathered in tables as the most important functions of this therapy.

## 2. OA and Pain

The pain in joints is recognized as a major and common symptom of all types of arthritis. According to different causes and symptoms of the disease, one can distinguish the various classes of arthritis and apply different treatment strategies [15]. Nevertheless, the pain associated with both RA/OA can have a determinantal impact on a patient’s life [16].

Cartilage degradation is mostly accompanied by severe pain. At the mechanism level, the pain in an advanced stage of OA is mainly caused by the infiltration of the patient’s synovium in the joints by macrophages. The activated macrophages then release pro-inflammatory cytokines, causing chronic pain and inflammation. From a diagnostic point of view, the level of the three pro-inflammatory mediators in the synovial tissue is significantly elevated, which can be considered as a diagnostic marker of OA. The main three types of pro-inflammatory mediators are (1) interleukin-1β (IL-1β), (2) IL-6, and (3) nerve growth factor (NGF). Therefore, these markers are currently considered promising clinical markers for the monitoring of OA progression [17].

IL-1 is a potent cytokine that can induce bone erosion in inflammatory sites, such as rheumatoid joint regions via activation of osteoclasts. Not only is IL-1 capable of activating osteoclasts, it is also a key cytokine involved in the differentiation, multinucleation, and survival of osteoclasts. IL-1 has two isoforms: IL-1α and IL-1β. Both isoforms bind to the same IL-1 receptor and have identical biological activity [18]. However, unlike IL-1α, IL-1β is secreted and circulates systemically, and is produced mainly by monocytes and macrophages. These two cytokines are also differentially regulated during development and in response to environmental signals, resulting in their different contributions to immune responses. Both are expressed at low levels under normal conditions and require induction at both the transcriptional and the translational levels [19]. From a diagnostic point of view, their processing and secretion have controlled mechanisms, and the loss of the regulatory stage leads to syndromes characterized by arthritis, fever, and rash [20].

The IL-6 is a proinflammatory cytokine that interacts in complex ways with cells involved in bone remodeling. In RA, IL-6 may indirectly promote osteoclastogenesis by increasing the release of RANK-L by osteoblasts, and it diminishes the proliferation of osteoblasts at late differentiation stages [21]. According to these facts, it plays a key role in the local and systemic manifestation of RA [22].

The NGF belongs to a family of proteins named neurotrophins, consisting of NGF, brain-derived neurotrophic factor (BDNF), neurotrophin-3 (NT-3), NT-4/5, and NT-6. NGF regulates a large number of physiological mechanisms that result in neurotrophic, metabotropic, and/or immunotropic effects [23]. A series of positive trials in the treatment of OA have provided an impetus to use anti-NGF as an analgesic [9]. However, there is a risk of the development of neuroarthropathy after using high doses of anti-NGF antibodies [24]. In addition, the regenerative capacity of the nervous system is limited, increasing preclinical evidence indicates that PRP would contribute to promoting the repair of this tissue [25].

## 3. The First-Line Therapy of Osteoarthritis

Nonpharmacological interventions, in parallel to the pharmacological treatment modalities, have been recommended for the management of OA and associated comorbidities [26]. The American Academy of Orthopaedic Surgeons (AAOS) published the guidelines to assist in making evidence-based decisions in the conservative management of hip and knee OA [27,28]. Moreover, other scientific societies, such as The European League Against Rheumatism (EULAR), the OsteoArthritis Research Society International (OARSI), and the American College of Rheumatology (ACR), have developed various recommendations for treating OA, depending on its location [2,29,30]. The pharmacological approach in symptomatic OA management mainly focuses on improvement or control of symptoms, pain, and underlying inflammation resulting in a reduced need for rescue analgesia and joint replacement surgery.

Before starting therapy, each patient should be assessed for medical conditions, such as hypertension, cardiovascular disease, heart failure, risk of gastrointestinal bleeding, chronic kidney disease, or other comorbidities, as well as trauma, disease severity, and surgical history that may affect the risk of side effects of certain pharmacological agents [2].

Drugs available for the treatment of OA (ATC code: M01-M09 [31]) include topical products for joint and muscular pain (ATC code: M02A [32]). The most frequently used preparations are the ATC subgroup M02AA [33], which includes anti-inflammatory, non-steroidal drugs (NSAIDs), e.g., those listed in Figure 1. Recently, pharmacological therapy is broadened by so called disease-modifying drugs for osteoarthritis (DMOADs) and symptomatic slow-acting drugs for osteoarthritis (SYSADOA) [34]. The initial oral medication of choice in the treatment of OA involves oral NSAIDs. Regardless, the anatomic location of pain must come before patients are recommended the available oral medications [2].

The ultimate goal for the OA treatment would be to halt the disease progression and repair the cartilage damage and relieve pain. However, effective cures are not currently available and recommendations for treatment vary. OA has multiple etiologies and interconnecting pathways; therefore, targeting only one molecule does not seem to become the effective treatment [5]. Due to this, it is highly unlikely to use a single medication effectively treating these diseases. OARSI and EULAR guidelines recommend that if patients do not respond to oral analgesics, they should receive intra-articular injections of either corticosteroids or hyaluronate followed by the use of opioids and narcotics only when all other pharmacological options have been considered [2,29,30,35].

### 3.1. Nonsteroidal Anti-Inflammatory Drugs (NSAIDs)

NSAIDs (Figure 1) exhibit anti-inflammatory and analgesic effects, and therefore are prescribed to control manifestations of the local inflammatory process. All NSAIDs are equally valuable when administered at high doses. However, they cannot prevent erosions or alter OA progression [36]. Their mechanism of action mainly includes inhibition of cyclooxygenase, resulting in the formation of prostaglandins inhibiting the inflammatory process. The major inconveniences of NSAIDs are related to their cardiovascular, nephrotoxic, and gastrointestinal adverse effects; thus, pharmacological guidelines for the management of OA recommend paracetamol intake of only up to 4 g/day [37]. It is an over-the-counter analgesic and antipyretic but the specific mechanism of its action is not fully elucidated [38]. Taken in therapeutic doses, paracetamol is remarkably well tolerated and metabolized, and the metabolic products are excreted in the urine [39]. Although paracetamol has minimal effects on symptoms, it is widely recommended as a first-line oral analgesic, presumably due to its availability, low cost, and assumed safety [14].

Salsalate (ATC code: N02BA06 [40]) is an NSAID assigned to a group of drugs called salicylates. Structurally, salsalate is a dimer of salicylic acid [41] and has anti-inflammatory, analgesic, and antipyretic actions similar to aspirin (ATC code: N02BA01 [40]). The anti-inflammatory and analgesic effects of salsalate are probably mediated by the inhibition of prostaglandin synthesis. Although available for several decades, salsalate is not commonly used. Current indications include treatment toward reduction of pain, swelling, and joint stiffness caused by OA, or other rheumatic disorders [42,43]. Salsalate can be used at a comparatively high dose because it is insoluble at gastric pH and is mostly absorbed in the alkaline environment of the intestines [44]. The full benefit of taking this medication for arthritis may be achieved up to two weeks after treatment. However, it can cause serious blood motion difficulties, such as a heart attack, especially if taken for long periods [45]. Recent reports suggest that high doses of salicylates show an anti-inflammatory effect through the inhibition of NF-κB activation in monocytic, lymphocytic, and endothelial cells [46]. The topical study has been demonstrated that salicylates inhibited IL-1β- and TNF-α-induced proinflammatory cytokine expression by blocking NF-κB activation in cells [47]. It has also been shown that the biotransformation of aspirin yields salicylate, a compound that possesses similar anti-inflammatory potency as the parent molecule [48].

If paracetamol cannot control symptoms or if inflammation signs are detected, the use of other NSAIDs is recommended by EULAR and OARSI at the lowest dose, with consideration of a gastroprotective agent [49]. Of course, it should be remembered that NSAIDs have serious toxic effects, including gastrointestinal irritation and ulceration, bleeding, and decreased renal blood flow with azotaemia. Patients taking anticoagulants who wish to take an NSAID should use a COX-2 inhibitor (such as celecoxib: COX-1/COX-2 as measured in vitro with human whole blood assays of a COX-isozyme activity, IC_50_ ratio = 30 [50]), which would not increase bleeding. According to the review of Puljak et al., there is evidence that celecoxib is slightly more efficient in pain reduction and conserves better joint function than other NSAIDs [51]. Furthermore, an analysis of 192 trials comprising more than 102,000 individuals revealed that etoricoxib and diclofenac are the most potent treatment options for OA, and the etoricoxib is safer than diclofenac [52]. Diclofenac is another good example of anti-inflammatory medicine dedicated to OA that predominantly inhibits COX-2 (COX-1/COX-2 as IC_50_ ratio = 29) [53].

Patients with dyspepsia should use proton pump inhibitors (PPI) and/or a COX-2 inhibitor. Patients with history of bleeding peptic ulcer are typically not prescribed NSAIDs at all [54]. Meloxicam (ATC code: M01AC06: [55]), a preferential inhibitor of COX-2 (COX-1/COX-2 as IC_50_ ratio = 18 [50]), is commonly used in the symptomatic treatment of OA. Human studies highlighted the effectiveness and tolerance of the meloxicam treatment in OA [56], comparable to that of diclofenac. Intra-articular meloxicam reduces cartilage matrix degeneration, and synovial inflammation in rat models of OA triggered chemically (by iodoacetate) [56] or by anterior cruciate ligament transection (ACLT) [57]. In humans, meloxicam reduces the Western Ontario and McMaster Universities Osteoarthritis Index (WOMAC) pain subscale, stiffness, and function subscale scores [58].

Alternative pharmacological therapy should be used only in the presence of inadequate response and severe pain [35]. As a prescription analgesic with an additional antipyretic activity, metamizole has an analgesic potency within the order of magnitude of a weakly effective opioid, such as tramadol, tilidine ATC code: N02AX [59] (Figure 2) or codeine.

OA is not an indication for metamizole per se, because it does not exert any anti-inflammatory effect. Metamizole is generally well tolerated, but in very rare cases serious life-threatening complications, such as agranulocytosis or allergic reactions, can occur, which in the worst case can culminate in anaphylactic shock [60].

Current therapies for OA focus on reducing pain, improving physical function, and quality of life in patients [61,62].

### 3.2. Disease Modifying Drugs for Osteoarthritis (DMOADs)

DMOAD is a class of medications aimed at affecting catabolic processes associated with OA pathogenesis, structural pathologies, and consequent clinical symptoms [63,64]. It should be emphasized here that regulators, such as the US Food and Drug Administration (FDA) and the European Medicines Agency (EMA), have not yet approved any drug as an effective DMOAD, as the approval guide requires a potential DMOAD to demonstrate a slowdown in knee joint width loss or hip on X-rays with symptomatic improvement. Such pharmaceuticals that will alter the natural history of disease progression by arresting the structural changes in the joints and relieving symptoms, either by reducing pain or improving physical function, are termed “DMOADs” [65]. The chemical structures of these drugs are presented in Figure 3.

Diacerein (Figure 3) is an anthraquinone synthesized in 1980 that interferes with interleukin-1, an inflammatory mediator. It is an NSAID with unique pharmacological properties as antioxidant and antiapoptotic, which has demonstrated great efficacy with the safety profile and SYSADOA. Its biological activities cover anti-inflammatory, anti-catabolic, pro-anabolic properties on cartilage and synovial membrane together with protective effects against subchondral bone remodeling [66]. Substantial effects against three types of pain have been reported for diacerein, namely antinociceptive effects [67,68,69], antineuropathic pain [70], and anti-inflammatory pain [71]. The mechanism by which diacerein inhibits inflammation and prevents cartilage ECM degradation is potentially associated with the inhibition of IL-1β converting enzyme, the reduction of IL-1β receptors on chondrocytes, and the increase in IL-1 receptor antagonists. Diacerein was also shown to affect matrix biosynthesis and subchondral bone remodeling in an arthritis sheep model [72].

Quercetin (Figure 3) has been shown to alleviate osteoarthritis by reducing inflammation and apoptosis in chondrocytes. In addition, it showed immunomodulatory effects in modulating the polarization of synovial macrophages to M2 macrophages to create a pro-chondrogenic microenvironment for chondrocytes, which in turn increased chondrocyte glycosaminoglycan synthesis and repaired damaged cartilage [73].

Recently, the Wing-like glycoproteins (Wnt) and transforming growth factor β (TGF-β) signaling pathways in the treatment of OA have been identified as a potential therapeutic target for DMOAD drugs. Wnt is defined as secreted glycoproteins that activate various intracellular signal transduction pathways and are involved in the differentiation and maturation of osteoblasts [74,75]. TGF-β is a theoretical target for the treatment of OA due to its capacity to protect cartilage under healthy conditions. It stimulates proteoglycan synthesis and inhibits hypertrophic chondrocyte differentiation [76]. However, it may be deleterious in aged individuals or in damaged cartilage [77]. Cell therapy approaches with partially transduced chondrocytes to express TGF- β have recently demonstrated statistically significant improved clinical scores [78]. Similar, the evidence for Wnt signaling involvement in cartilage homeostasis had been established [75].

Crosstalk between TGF-β and Wnt signaling was demonstrated in chondrocytes and revealed that TGF-β activates β-catenin signaling via SMAD3 and β-catenin interaction. Later, the mechanism of action showed that both SMAD3 and SMAD4 were required for the interaction with β-catenin, thereby providing protection from proteasomal degradation [79].

In this group of mechanisms, two compounds are currently recognized as the most promising candidate drugs i.e., sprifermin and lorecivivint. However, in the development of the conducted studies, it is difficult to clearly predict their clinical application [63].

Sprifermin is a potential DMOAD that has been shown to increase articular cartilage thickness compared with a placebo in patients with knee OA at the primary endpoint of the FORWARD trial and was well tolerated [80]. Structural benefits with sprifermin were observed in a subgroup at risk [64].

Lorecivivint (SM04690, Figure 3) is a small-molecule Wnt pathway modulator currently in development as a potential DMOAD for the treatment of knee OA [81]. It modulates the Wnt signaling pathway by inhibiting CDC-like kinase 2 and dual-specificity tyrosine phosphorylation-regulated kinase 1 A which are molecular regulators in Wnt signaling, chondrogenesis, and inflammation [82].

Most clinical trials investigating the disease-modifying effects demonstrated by DMOAD prodrugs are the story of failure. For example, inhibitors of IL-1, IL-6 or TNF-α in OA patients failed to meet the primary and secondary endpoints, such as in cases of gevokizumab (XOMA-052) [65], AMG108 [83], lutikizumab (ABT-981) [84,85], tocilizumab [86], anakinra [87], adalimumab [88], and etanercept [89]. These highlights and proves the complexity and difficulty of the challenges faced by many researchers who tirelessly seek the appropriate therapy for osteoarthritis [90,91]. It also answers the question as toa why there is no DMOAD agent approved by a regulatory agency to the date.

### 3.3. Symptomatic Slow-Acting Drugs for Osteoarthritis (SYSADOA)

Small molecular chemical compounds are an abundant source of new pharmaceutical therapeutics for cartilage regeneration, as they have advantages in design, fabrication, and application [72].

The most popular SYSADOAs are glucosamine and chondroitin (Figure 3). Based on the results of randomized controlled trials, it is considered that consumption of chondroitin for certain dosage has a positive effect on pain relief and improvement in function [92]. A multicenter, randomized clinical trial demonstrated a lack of superiority of chondroitin and glucosamine combination therapy over the placebo in a six month follow-up [93]. Although some studies have shown significant safety and efficiency, there is still controversy regarding the superiority over the placebo. International guidelines for the management of OA had given an equivocal recommendation of glucosamine and chondroitin, and they are not recommended [94].

### 3.4. Glucorticosteroid Therapy and Viscosuplemmentation

Patients unable to take NSAIDs or who do not respond to NSAIDs can be given intra-articular corticosteroid injections [54]. Glucorticosteroid (Figure 4) therapy is the oldest and best-known intra-articular treatment.

Despite controversy about the long-term effect of GS on articular cartilage, it has been shown to be highly effective in short-term pain relief and functional improvement in patients with OA and a relatively low risk of side effects [95]. A systematic review of the use of steroid injections found that the beneficial effect of treatment started 1 week after the injection and lasted 3–4 weeks. Other authors have confirmed these results, noting some long-term benefits up to 6 months after administration in knee OA [96]. A systematic review of the Cochrane database, including the results of 28 studies comparing steroid injections with a placebo showed that GS is more effective in reducing pain and improving the patient’s overall self-esteem 3 weeks after injection [97]. The authors concluded that intra-articular steroid injections rapidly reduce the symptoms of OA, but the therapeutic effect is short-term. The mechanism by which GS applied to the degenerated knee joint reduces the inflammatory cascade is not fully understood. This is likely to happen by reducing the pro-inflammatory effects of arachidonic acid. Direct action on nuclear steroid receptors leads to changes in the functions of T and B lymphocytes, a decrease in the level of cytokines, and inhibition of phospholipase A2, resulting in a reduction of pro-inflammatory arachidonic acid derivatives [98]. The greatest concern during therapy is related to the potential side effects. The most important of these is the harmful effect of long-term steroid use on the tissues of the joint. On the other hand, the most frequently reported complications are skin atrophy at the injection site and facial flushing within 24–48 h after injection [99]. Systemic side effects, such as joint infection, increased blood glucose, and cortisol levels, are rarely observed. This is particularly important in diabetic patients with a rapid increase in blood glucose levels 2–3 days after the injection of GS into the knee joint [100]. The results of preclinical studies suggest that GS may be detrimental to the viability of chondrocytes [101]. The chondrotoxic effect depends primarily on the dose used and the frequency of administration. In the case of betamethasone, doses above 2.1 mg negatively affect chondrocytes, interfering with protein synthesis and contributing to the breakdown of cartilage. On the other hand, lower doses (<2.1 mg) have a chondroprotective effect by reducing inflammation [102]. Another subject of controversy related to intra-articular GS is the potential risk of immunosuppression. Several studies have found an increased risk of perioperative infection following knee replacement surgery after prior intra-articular steroid injections [103,104,105]. There is no consensus on how long a break should be taken to avoid this dangerous complication, but some authors recommend a minimum 3-month period of refraining from GS injections before surgery [106].

Hyaluronic acid (Figure 4, ATC code: S01KA01) is a naturally occurring component of cartilage and synovial fluid. Responsible for the rheological properties of the synovial fluid, reducing the friction between the articular surfaces, and ensuring the amortization of loads. It is a polysaccharide from the group of glycosaminoglycans. Its molecular weight in normal synovial fluid ranges from 6500 to 10,900 kDa. A normal adult knee has about 2–4 mL of synovial fluid, and the concentration of hyaluronic acid ranges from 2.5 to 4.0 mg/mL [107]. In a healthy joint, the average half-life of HA is approximately 20 h. In OA, the half-life is almost 50% less (to 11–12 h) [108]. In addition, OA reduces the viscoelastic properties of HA.

The purpose of intra-articular injections of HA in OA is primarily to improve these viscoelastic properties. For this reason, this form of therapy is sometimes referred to as “viscosupplementation”. HA also exerts anti-inflammatory, analgesic, and possibly chondroprotective effects [107]. The relatively long period of clinical improvement after viscosupplementation is also because restoring joint homeostasis by administering exogenous HA leads to an increase in its endogenous production [109].

Currently, many preparations of HA are available with different molecular weights (from 500 to 6000 kDa). There are two main production technologies: synthesis by bacterial fermentation or extraction from animal tissues, e.g., rooster combs. Despite the great variety, the superiority of any preparation over other brands has not yet been proven. There is also no clear evidence supporting specific clinical criteria for selecting patients who are likely to improve after injection. In a meta-analysis of controlled clinical trials, Wang et al. showed that patients in late-stage OA and the elderly (>65 years of age) have a lower chance of reducing symptoms than younger patients or patients in the early stage of OA [108]. HA therapy is contraindicated in people with hypersensitivity to hyaluronic products, pregnant or lactating women, pediatric patients, and patients with an infection within the treated knee joint.

## 4. Platelet-Rich Plasma (PRP)/Platelet-Rich Fibrin (PRF) as Second Line Treatment

Platelets (PLT), also known as thrombocytes, are small cytoplasmic fragments (2 to 3 μm in diameter, volume between 5 and 10 fL (10^−15^ L (femtoliters) and the normal counts in peripheral blood (approximately 150–450 × 10^9^ per liter volume of plasma)) [110]). They are derived from bone marrow megakaryocytes. Numerous studies have established that they consist of more than 800 proteins and other molecules. These components are GFs (also known as the osteoinductive proteins), cytokines, chemokines, membrane proteins, metabolites, messenger molecules, and numerous soluble proteins [111].

PRP is defined as a volume of plasma with a platelet concentration (1 × 10^6^ PLT/μL) in small volume of plasma (approximately 5 mL) higher than the average in peripheral blood [111]. According to this definition, PRP contains 2–9 folds more PLT per unit volume than normal blood. PRP is developed from autologous blood, it is inherently safe, and is free from transmissible diseases. It has been used successfully on tissue regeneration, due to the high concentration of GF contained in platelets [112]. GFs, which are being released from platelets α-granules enhance the regeneration of injured tissue. The α-granules are approximately 200 to 500 nm in diameter and contain over 30 bioactive proteins [113].

A review of the literature shows that the history of various PRP-based therapies is at least 70 years old. For this reason, it is widely accepted that the method is sufficiently well-established to allow widespread use in medicine, particularly in sports medicine, and orthopedics [114]. The use of PRP in the process of plasmapheresis in enhancing bone regeneration was described in 1999 by Anitua [115,116]. A recent meta-analysis of randomized controlled trials (RCT) evaluating the effects of PRP or intra-articular hyaluronic acid (IA HA) injections in patients with knee OA documented that PRP is an effective treatment that reduced pain and improved functional status [117,118].

In sum, PRP is an autologous orthobiologic treatment option for musculoskeletal conditions with favorable results in a limited number of high-quality clinical trials. PRP therapy is currently used to treat degenerative disorders and promote the healing of tendons, ligaments, muscles, and cartilage [114].

### 4.1. PRP as Source of GF

PRP includes 600 GFs [119,120] and the most important GFs are: platelet-derived growth factor (PDGF, consisting of two dimeric isoforms: PDGFαα, PDGFββ, and heterodimer PDGFαβ [121]; TGF-β, including TGF-β1-3 [122], vascular endothelial growth factor (VEGF) [123]; epidermal growth factor (EGF) [124]; fibroblast growth factor (FGF); insulin-like growth factor (IGF, including IGF-1 and IGF-2) [125,126]; hepatocyte growth factor (HGF) [127,128]; connective tissue growth factor (CTGF) [129]; bone morphogenetic proteins (BMPs) [130,131]; and nerve growth factor (NGF). In Table 1, the most crucial GFs in PRP and their functions are listed.

Essentially, GFs perform their function by binding ligands to related extracellular receptors on target cells, leading to the binding of intracellular cytoplasmic proteins to phosphorylated tyrosine. It is a complex process that stimulates or inhibits cell migration, proliferation, and differentiation [155].

The PRP has been widely used in stomatology, orthopedics, and recently in sports medicine and aesthetic medicine/plastic surgery. The rationale for the clinical use of PRP is based on its ability to stimulate the production and, accordingly, the increase in the concentration of growth factors and in the secretion of proteins, which are able to maximize the healing process at the cellular level [125,157].

VEGF is the master regulator of angiogenesis in physiology and disease. In addition, GFs are probably able to influence the neovascularization process associated with degenerative changes [158]. The exploitation of its therapeutic potential, though, requires careful control of its dose distribution within tissue. Co-delivery of platelet-derived growth factor-BB (PDGF-BB) has been shown to expand the therapeutic window of VEGF and also improve associated arteriogenesis. The potential effect of PRP on neovessels has already been extensively studied in Achilles tendinopathy [157,159].

### 4.2. PRP as Source of Cytokines

Cytokines are bioactive proteins produced by many different cells of the immune system, such as neutrophils, monocytes, macrophages, B-cells, and T-cells [160]. Cytokines differ not only in their function, but also have a wide variety of molecular weight ranges (approximately 6–70 kDa). There are several different families of cytokine proteins and the number of identified proteins continues to grow [160]. The cytokines present in PRP are IL (interleukin family: IL-1β, IL-4, IL-6, IL-8, IL-10, and IL-13), IP-10 (interferon-γ-inducible protein 10, CXCL10), TNF-α (tumor necrosis factor-α), MMP-9 (matrix metalloproteinase-9), and NF-κB (kappa-light-chain-enhancer of activated B cells) [135]. The most important cytokines in PRP and their functions are gathered in Table 2.

Arthritis is an inflammatory disease of diarthrodial joints and is associated with swollen inflamed joints, disruption of joints, and loss of integrity of articular cartilage and synovial joints [15]. It is well-known that cytokines play an important role in inflammation and immune response. The inflammatory mediators can be detected in both synovial fluid and serum in OA patients, indicating that inflammation plays a significant role in the pathogenesis of OA [162]. This makes them therapeutic targets in chronic arthritis, with TNF-α and IL-1 as main focuses [181]. The inflammatory phase of bone healing is regulated by proinflammatory cytokines secreted by invading macrophages, polymorphonuclear leukocytes, and lymphocytes [155].

The production of these enzymes is controlled by NF-κB. Although PRP does not act directly on NF-κB, the factors contained in alpha granules and released with thrombocyte activation may counteract the effects of PRP-induced cartilage degradation [183,184]. Lee et al. reported a decrease in inflammation in the joint after administration of PRP, as well as an increase in the level of messenger RNA for cannabinoid receptors, i.e., CB1 and CB2 [185]. Thus, in addition to its anti-inflammatory effects, PRP may also have potential analgesic effects. Experimental studies have shown that in OA, PRP also affects synoviocytes, endothelial cells, and cells involved in cellular immunity (e.g., macrophages) [186,187]. Additionally, it has been shown that the concentration of growth factors, such as PDGF, β-FGF, VEGF, and IGF-1 increases at various time points within 96 h after the injection of PRP. The results of these studies suggest that PRP not only provides growth factors but also activates biological processes leading to their release [188].

Research to date shows that intra-articular or intra-osseus injection of PRP is both an effective and safe treatment for OA [189]. However, few randomized clinical trials have been conducted to assess its greater effectiveness concerning other, less expensive methods of conservative treatment. In addition, several aspects, such as the duration of treatment, the optimal frequency, and the interval between injections, require further research. In 2017, members of the AAOS published the Minimum Information for Studies Evaluating Biologics in Orthopedics (MIBO) guidelines for the design and reporting of studies using PRP and mesenchymal stem cells [190]. According to these guidelines, research should focus on more homogeneous patient groups, standardizing the protocols for PRP administration and post-injection management. In the future, the identification of early OA biomarkers and new imaging methods of the knee may result in a more accurate assessment of the effectiveness of PRP therapy, as well as help determine the appropriate dosing schedule. The method of storage and transport is also important in clinical trials and should be described in publications. The combination of PRP and HA seems to be a reasonable solution in mild to moderate OA treatment [191]. A meta-analysis comprising 337 patients comparing HA-PRP injections and HA alone concluded that for symptomatic patients with knee OA, the combination of PRP and HA was more effective in analgetic and functional improvement compared to patients who received HA injections only, as assessed by 3-, 6-, and 12-month VAS scores, and 12-month WOMAC physical function and stiffness scores [192]. Nowadays, there is interest in innovative and efficient cellular therapies that also act due to GFs. The autologous micro-fragmented adipose tissue (MFAT) can promote the regeneration of degenerated chondrocytes and improvement of the subchondral bone lesions in animal models [193]. Moreover, in human studies, MFAT injections improved clinical outcomes in mild knee osteoarthritis treatment [194]. MFAT can be considered as a stem cell-based natural scaffold for cartilage defect repair. A combination of MFAT and PRP seems to be reasonable due to the presented advantages. Nevertheless, harvesting and processing techniques are vital to achieving optimal MSCs and growth factors for satisfactory results.

## 5. Conclusions

Research into the pathogenesis of OA has led to the development of a variety of pharmacological products. Pain medications, including the most popular NSAIDs, are the first-line treatment. Local treatment has been used increasingly more widely, which in the hierarchy of proceedings, takes a place between rehabilitation and pharmacological treatment. Only in the case of no improvement and the presence of advanced lesions visible in imaging tests, should surgery be considered. The use of topical therapy in the treatment of osteoarthritis has the advantage over pharmacological treatment in that it can significantly alleviate the symptoms of OA while avoiding the side effects associated with systemic therapies, such as gastrointestinal bleeding and cardiovascular complications. Currently, an increasing number of published studies suggest that intra-articular injections may be as effective or even more effective than NSAIDs and result in fewer systemic adverse events. The growing popularity of intra-articular injections in OA is associated with their effectiveness, a low number of complications, and a simple injection technique performed on an outpatient basis.

However, the major challenges in OA treatment are to understand the mechanism of action in the joint to optimize and standardize the dosing schedule, identify the most suitable biomarkers for treatment efficacy assessment, and reveal the underlying mechanisms involved in OA pathophysiology. Most of the actual treatments succeed in resolving the pain and providing better function, but they are unable to permanently stop the degenerative process. Various experimental and clinical studies conducted to date have used intra-articular injections and have been able to demonstrate the positive effect on structural modulation and anti-inflammatory effects in OA. The dosing schedule and the variations in the intra-articular products are also often debated due to the large number and variability of commercially available systems. The use of biological products is a rapidly growing field. Despite their wide clinical use, some of these products have been studied without rigorous scientific standards. This raises justified doubts as to the validity of the obtained therapeutic conclusions, both for clinicians and patients.

## Figures and Tables

**Figure 1 ijms-23-01566-f001:**
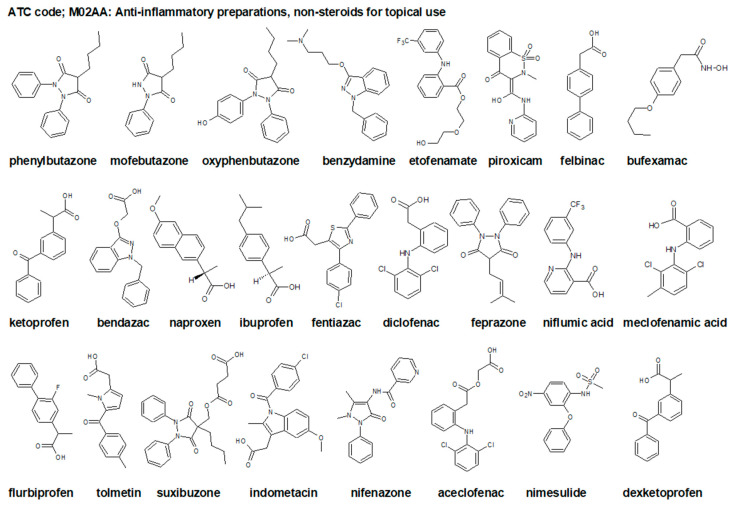
Chemical structures of selected non-steroidal anti-inflammatory drugs (NSAIDs), topical drugs for joint and muscle pain ATC code: M02AA [33].

**Figure 2 ijms-23-01566-f002:**
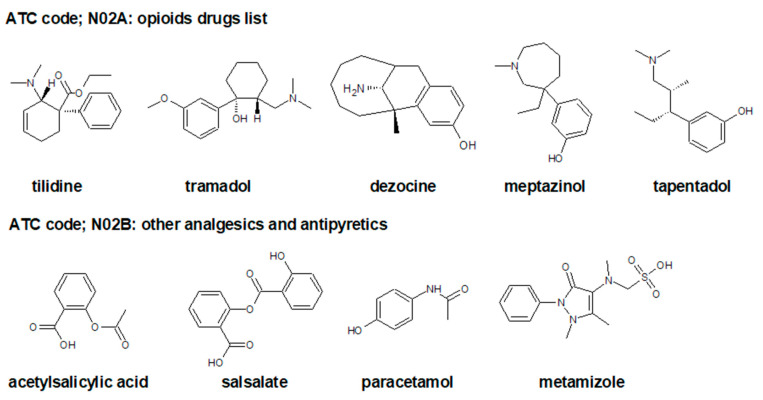
Chemical structures of selected analgesic drugs, such as opioid drug ATC code: N02A and other analgesic and antipyretics ATC code: N02B [59].

**Figure 3 ijms-23-01566-f003:**
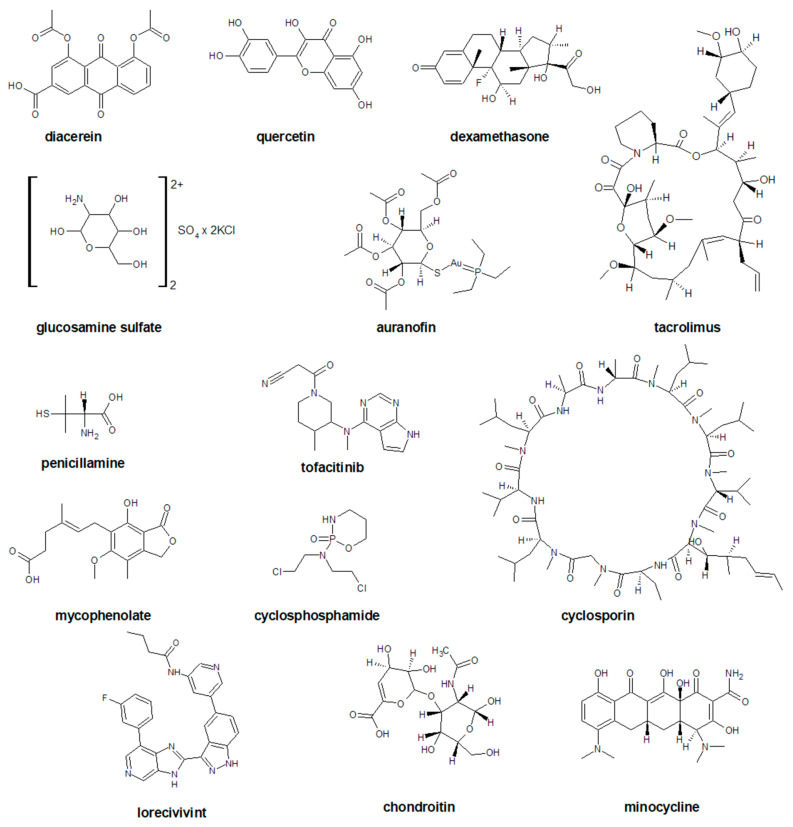
The chemical structures of selected disease modifying drugs for osteoarthritis (DMOADs).

**Figure 4 ijms-23-01566-f004:**
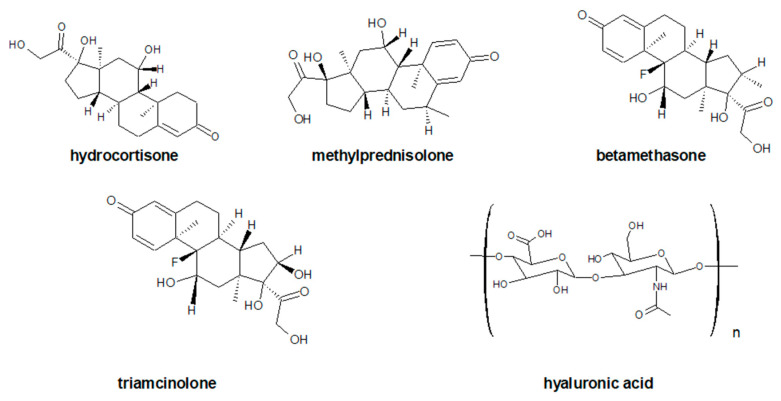
The chemical structures of selected glucocorticosteroids.

**Table 1 ijms-23-01566-t001:** Growth factors present in PRP and their functions.

Growth Factors/ References	Functions
PDGF [114,132,133,134,135,136,137]	The first growth factor to be approved for the treatment of human ulcers; Plays a significant role in blood vessel formation (angiogenesis process); Enhances proliferation of fibroblasts and their effectiveness in the production of extracellular matrix; Regulates cell growth and division; Helps in tissue remodeling and cellular differentiation; Helps in mesenchymal proliferation and directs the migration and differentiation; Promotes the healing of wounds.
TGF-β [12,122,138,139,140]	Controls proliferation process; Plays an important role in embryonic development, cell growth, cellular differentiation, hormone secretion, and immune function; Inhibits collagen synthesis and calcium release; An inducer of extracellular matrix proteins; Contributes to the reproductive process by enhancing endometrial proimplantatory LIF (Leukemia Inhibitory Factor).
VEGF [119,123,141,142,143,144]	Stimulates angiogenesis and vasculogenesis; Stimulates endothelial degradation and helps in the underlying basement membrane, migration into the surrounding tissue, proliferation and tube formation; Activates the synthesis of metalloproteinase; Initiates a macrophage-related angiogenic response in the inflammation stage; Important for the recruitment of blood vessels and osteoclasts in endochondral bone repair; Required for angiogenesis-osteogenesis coupling in intramembranous bone repair; Regulates osteoclasts in the remodeling stage; Useful in the treatment of impaired bone healing/regeneration.
EGF [124,145]	A protein that stimulates cell growth and differentiation by binding to its receptor (EGFR); Plays a substantial part in the maintenance of spermatogenesis.
FGF [120,146,147]	FGF-2, FGF-7, and FGF-10—crucial for wound healing; Involved in angiogenesis, wound healing, embryonic development, and various endocrine signaling pathways; Stimulates proliferation of various cells of mesodermal, ectodermal, and also endodermal origin; Helps in the regulation of many developmental processes, including morphogenesis, differentiation, cell proliferation, and migration during embryonic development.
IGF [126,148]	Growth hormone and its downstream mediator IGF construct a pleiotropic axis affecting growth, metabolism, and organ function; Exerts their effects on osteogenic cells via binding to their cognate receptor, leading to activation of an array of genes that mediate cellular differentiation and function; Interacts with skeletal regulators, such as sex steroids, thyroid hormone, and parathyroid hormone, to facilitate skeletal growth and metabolism; IGFs are produced in the liver; it bears on various reproductive processes and receives an essential role in the regulation of spermatogenesis; Regulates linear and radial bone growth and modeling during development; Protects the skeleton from age-related bone loss.
HGF [127,149,150]	Stimulates angiogenesis; Mitogenic for hepatocytes and endothelial cells; the key factor for tissue self-repair in animal models; protect liver, kidneys, lungs, heart, brain, and gastrointestinal mucosa, due to anti-apoptotic; and anti-inflammatory properties (as demonstrated on animal models); HGF neutralization at embryonic stage leads to hypoplasia; Cytotoxic effect on some cancer cells; One of the glucose controlling factors—preventive against hyperglycemia, stimulates insulin secretion Regulated osteoclastogenesis by JNK and AKT-GSK-3β-NFATc1 signalings.
CTGF [151,152,153]	Plays an important role in cell adhesion, migration, proliferation, angiogenesis, skeletal development and tissue repair; Stimulates cartilage regeneration; Promotes matrix mineralization Promotes the proliferation and differentiation of growth-plate chondrocytes, osteoblasts, and vascular endothelial cells, all of which play important roles in endochondral ossification; Promotes the formation of osteoclasts in vitro, which cells play an important role in the replacement of cartilage by bone during endochondral ossification; Promotes intramembranous ossification, regenerate cartilage and bone, and induce angiogenesis in vivo.
BMPs [131,154,155]	At least 20 BMPs have been identified; First discovered as proteins that induce ectopic bone formation; Known to be potent osteoinductive proteins involved in many of the processes related to bone formation and regeneration; Have unique activity in the demineralized bone matrix, which induces heterotopic bone formation in nonskeletal tissues, such as skeletal muscle and subcutaneous tissue; regulating cellular lineage commitment, morphogenesis, differentiation, proliferation, and apoptosis of various types of cells throughout the body; BMP-2 and BMP-7 are approved for clinical use in open fractures of long bones, non-unions, and spinal fusion.
NGF [9,156]	Ubiquitous seminal plasma protein in mammals; Substance is involved in regulating neuronal survival and differentiation; Acts as a pain mediator in mature mammals; Limits the regenerative capacity of the nervous system.

**Table 2 ijms-23-01566-t002:** Cytokines presents in PRP and their functions.

Cytokines/Reference	Functions
IL-1 [161,162,163]	Proinflammatory cytokine and pain mediator that acts as an endogenous pyrogen; Resulting in pain sensitization, bone resorption, and cartilage destruction; Family members include IL-1α, IL-1β, and IL-1ra (endogenous IL-1 receptor antagonist); Principal targets are macrophages, large granular lymphocytes, B cells, endothelium, fibroblasts, and astrocytes secrete IL-1; Acts on the hypothalamus to induce fever (also called an endogenous pyrogen); Operates on hepatocytes to increase the synthesis of specific serum proteins, such as amyloid A protein and fibrinogen; Causes fall in blood pressure or shock in large amounts; Causes lymphocyte activation, macrophage stimulation, increased leukocyte/endothelial adhesion, fever due to hypothalamus stimulation, and release of acute-phase proteins by the liver; Causes apoptosis in many cell types and cachexia; IL-1β plays important roles in inflammation and matrix degradation and is the primary target for reducing inflammation by manipulating IL-1ra; Corticosteroids inhibit the IL-1 effect.
IL-4 [164,165,166,167]	Intracellular anti-inflammatory cytokines; Specific cytokine modulating osteogenic capability; Primary effects include B-cell differentiation and stimulation of acute-phase proteins; Promotes mast cell proliferation in vivo; Promotes bone regeneration by causing the polarization of macrophage phenotype towards the anti-inflammatory M2, rather than the inflammatory M1 phenotype what promotes osteogenesis; Inhibits IFN gamma-mediated activation on macrophages; Promotes change in the cytokine microenvironment result in decreased osteoclast activity and increased osteoblast activity, resulting in the repair of bone Increases bone mineralization.
IL-6 [21,22,168,169]	Intracellular pro-inflammatory cytokines; The central mediators of osteoclast activity; appears, in part, to mediate the bone resorption–inducing effects of TNF-a and IL-l and osteoprotegerin (OPG); B lymphocytes and hepatocytes are its principal targets; Primary effects include B-cell differentiation and stimulation of acute-phase proteins.
IL-8 [170,171,172,173]	Stimulation of angiogenesis; Contributed to osteoclast formation; Principal targets are neutrophils, basophils, mast cells, macrophages, and keratinocytes; Plays a major part in infertility and is involved in mitogenesis, inhibition of angiogenesis, inflammation, chemotaxis, neutrophil degranulation, leukocyte activation, and calcium homeostasis.
IL-10 [161,174,175]	Intracellular anti-inflammatory cytokines; Principal targets are Th1 cells; Causes inhibition of IL-2 and interferon gamma; Decreases the antigen presentation, and MHC class II expression of dendritic cells; Costimulatory molecules on macrophages; Downregulates pathogenic Th17 cell responses; Inhibits the expression of many proinflammatory cytokines, chemokines, and chemokine receptors58 and mediates allergen tolerance in allergen-specific immunotherapy and after exposure to high doses of the allergen; Inhibits IL-12 production by macrophages; Promotes survival, proliferation, and differentiation of human B cells and increases the production of IgG4.
IL-13 [176,177]	Intracellular anti-inflammatory cytokines; Acts on monocytes, fibroblasts, epithelial cells and B cells; Significant effects are B-cell growth and differentiation, stimulates isotype switching to IgE. Causes increased mucus production by epithelial cells, increased collagen synthesis by fibroblasts and inhibits pro-inflammatory cytokine production; Works together with IL-4 in producing biologic effects associated with allergic inflammation and in defense against parasites.
IP-10 [174,175,178,179]	Proinflammatory cytokine; stimulate angiogenesis; Regulating osteoblast and osteoclast functions.
TNF-α [180,181,182]	Intracellular anti-inflammatory cytokines; Inflammation mediator; Stimulates the growth of fibroblasts; Promotes angiogenesis.

## Data Availability

Not applicable.

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
