# Peer review of "Overview of First-Line and Second-Line Pharmacotherapies for Osteoarthritis with Special Focus on Intra-Articular Treatment"

_ijms, 2022, doi:10.3390/ijms23031566_

Round 1

Reviewer 1 Report

Alicja Nowaczyk et al. released a comprehensive review of osteoarthritis's first-line and second-line therapies. The authors deal in detail with the main NSAIDs, disease-modifying drugs, symptomatic slow-acting drugs. A particular view on the effects of diacerein, quercetin, and the Wnt –inhibitors Sprifermin and Lorecivivint is offered. Finally, the composition, biological effects, and leading clinical results of platelet-rich plasma treatment are described.

This is a well-wrought review with important synthetic value. However, to reach the publication standards, several issues need to be improved, as follows:

  1. The title: the authors deal with non-cellular and non-biological treatments. They should stipulate this in the abstract. Therefore, I suggest a slight modification: "Overview of first-line and second-line pharmacotherapies for osteoarthritis with special focus on intra-articular treatment."

  1. Platelet-reach plasma is usually abbreviated as PRP, not PPR. Consensus abbreviations are important for searching engines.
  2. There is incomplete information concerning the DMOADS in this review. The references also should be completed.
  3. A disease-modifying drug candidate is the Cox-2 preferential inhibitor, Meloxicam. Intra-articular Meloxicam reduced cartilage matrix degeneration, and synovial inflammation in rat models of OA triggered chemically (by iodoacetate) [Csifo et al.] or by anterior cruciate ligament transection (ACLT) [Wen et al.]. In humans, Meloxicam reduced the WOMAC pain subscale, stiffness, and function subscale scores [Altman et al.].
  4. It would also be worthwhile to oppose the Cox-2 selective celecoxib with other NSAIDs. According to the review of Puljak et al., there is evidence that celecoxib is slightly better in reducing pain and conserves better joint function than other NSAIDS [Puljak et al.]. Furthermore, an analysis of 192 trials comprising more than 102.000 individuals revealed that etoricoxib and diclofenac are the most potent treatment options for OA, and the etoricoxib is safer than diclofenac [da Costa et al.].
  5. The authors present in detail the benefits of PRP. However, other autologous biological therapies also act due to growth factors. Innovative and efficient treatment is the autologous micro-fragmented adipose tissue local injection. MFAT promoted the regeneration of cartilage, migration capacity of chondrocytes, and improvement of the subchondral bone lesions in animal models (Xu ). Moreover, in human studies, MFAT improved the Tegner Lysholm Knee, VAS, IKDC-subjective, and the total KOOS scores (Russo). MFAT supplies mesenchymal stem cells, biological scaffolds, and growth factors to the joint. I suggest mentioning this therapeutical option, too.
  6. There are in the manuscript some improper citations, like:

“A recent meta-analysis of randomized controlled trials (RCT) evaluating the effects of PPR or intra-articular hyaluronic acid (IA HA) injections in patients with knee OA documented that PRP is an effective treatment that reduced pain and improved functional status [35].” Please, correct them.

  1. Concerning the OARSI treatment guidelines, the newest version should be included among the references [Bannuru et. al.]

If these improvements are performed, I agree with the publication of this manuscript in IJMS.

References

  1. Altman R, Hochberg M, Gibofsky A, Jaros M, Young C. Efficacy and safety of low-dose SoluMatrix meloxicam in the treatment of osteoarthritis pain: a 12-week, phase 3 study. Curr Med Res Opin. 2015 Dec;31(12):2331-43. doi: 10.1185/03007995.2015.1112772. PMID: 26503347.
  2. Bannuru RR, Osani MC, Vaysbrot EE, Arden NK, Bennell K, Bierma-Zeinstra SMA, Kraus VB, Lohmander LS, Abbott JH, Bhandari M, Blanco FJ, Espinosa R, Haugen IK, Lin J, Mandl LA, Moilanen E, Nakamura N, Snyder-Mackler L, Trojian T, Underwood M, McAlindon TE. OARSI guidelines for the non-surgical management of knee, hip, and polyarticular osteoarthritis. Osteoarthritis Cartilage. 2019 Nov;27(11):1578-1589. doi: 10.1016/j.joca.2019.06.011. Epub 2019 Jul 3. PMID: 31278997.
  3. Csifó (Vajda), E.; Nagy, E.E.; Horváth, E.; Fárr, A.-M.; Muntean, D.-L. Mid-term effects of meloxicam on

collagen type II degradation in a rat osteoarthritis model induced by iodoacetate. Farmacia 2015, 63, 556–560

  1. da Costa BR, Pereira TV, Saadat P, Rudnicki M, Iskander SM, Bodmer NS, Bobos P, Gao L, Kiyomoto HD, Montezuma T, Almeida MO, Cheng PS, Hincapié CA, Hari R, Sutton AJ, Tugwell P, Hawker GA, Jüni P. Effectiveness and safety of non-steroidal anti-inflammatory drugs and opioid treatment for knee and hip osteoarthritis: network meta-analysis. BMJ. 2021 Oct 12;375:n2321. doi: 10.1136/bmj.n2321. PMID: 34642179; PMCID: PMC8506236.
  2. Nagy E, Vajda E, Vari C, Sipka S, Fárr AM, Horváth E. Meloxicam ameliorates the cartilage and subchondral bone deterioration in monoiodoacetate-induced rat osteoarthritis. PeerJ. 2017;5:e3185. Published 2017 Apr 12. doi:10.7717/peerj.3185
  3. Puljak L, Marin A, Vrdoljak D, Markotic F, Utrobicic A, Tugwell P. Celecoxib for osteoarthritis. Cochrane Database Syst Rev. 2017;5(5):CD009865. Published 2017 May 22. doi:10.1002/14651858.CD009865.pub2
  4. Russo A, Screpis D, Di Donato SL, Bonetti S, Piovan G, Zorzi C. Autologous micro-fragmented adipose tissue for the treatment of diffuse degenerative knee osteoarthritis: an update at 3 year follow-up. J Exp Orthop. 2018 Dec 19;5(1):52. doi: 10.1186/s40634-018-0169-x. PMID: 30569417; PMCID: PMC6300453.
  5. Wen ZH, Tang CC, Chang YC, Huang SY, Chen CH, Wu SC, Hsieh SP, Hsieh CS, Wang KY, Lin SY, Lee HL, Lee CH, Kuo HC, Chen WF, Jean YH. Intra-articular injection of the selective cyclooxygenase-2 inhibitor meloxicam (Mobic) reduces experimental osteoarthritis and nociception in rats. Osteoarthritis Cartilage. 2013 Dec;21(12):1976-86. doi: 10.1016/j.joca.2013.09.005. Epub 2013 Sep 29. PMID: 24084190.
  6. Xu T, Yu X, Yang Q, Liu X, Fang J, Dai X. Autologous Micro-Fragmented Adipose Tissue as Stem Cell-Based Natural Scaffold for Cartilage Defect Repair. Cell Transplant. 2019 Dec;28(12):1709-1720. doi: 10.1177/0963689719880527. Epub 2019 Sep 29. PMID: 31565996; PMCID: PMC6923561.

Author Response

Reviewer #1: Comments to the Author

(x) English language and style are fine/minor spell check required.

Authors’ answer:

The language of the manuscript was improved and checked by native English speaker. All changes are indicated in the manuscript. Thank you very much for this comment.

R#1: Alicja Nowaczyk et al. released a comprehensive review of osteoarthritis's first-line and second-line therapies. The authors deal in detail with the main NSAIDs, disease-modifying drugs, symptomatic slow-acting drugs. A particular view on the effects of diacerein, quercetin, and the Wnt –inhibitors Sprifermin and Lorecivivint is offered. Finally, the composition, biological effects, and leading clinical results of platelet-rich plasma treatment are described.

This is a well-wrought review with important synthetic value. However, to reach the publication standards, several issues need to be improved, as follows:

  1. The title: the authors deal with non-cellular and non-biological treatments. They should stipulate this in the abstract. Therefore, I suggest a slight modification: "Overview of first-line and second-line pharmacotherapies for osteoarthritis with special focus on intra-articular treatment."

Authors’ answer:

According to referee's suggestions we have modified the title. Thank you for this valuable comment.

R#1:

  1. Platelet-reach plasma is usually abbreviated as PRP, not PPR. Consensus abbreviations are important for searching engines.

Authors’ answer:

Thank you very much for that important suggestion! Corresponding corrections were made in the new version of the manuscript.

R#1:

  1. There is incomplete information concerning the DMOADS in this review. The references also should be completed.

Authors’ answer:

Thank you for suggestion – we extended this section to show hot topic in treatment with described therapeutic options.

R#1:

  1. A disease-modifying drug candidate is the Cox-2 preferential inhibitor, Meloxicam. Intra-articular Meloxicam reduced cartilage matrix degeneration, and synovial inflammation in rat models of OA triggered chemically (by iodoacetate) [Csifo et al.] or by anterior cruciate ligament transection (ACLT) [Wen et al.]. In humans, Meloxicam reduced the WOMAC pain subscale, stiffness, and function subscale scores [Altman et al.].

Authors’ answer:

Thank you for this valuable comment.

R#1:

  1. It would also be worthwhile to oppose the Cox-2 selective celecoxib with other NSAIDs. According to the review of Puljak et al., there is evidence that celecoxib is slightly better in reducing pain and conserves better joint function than other NSAIDS [Puljak et al.]. Furthermore, an analysis of 192 trials comprising more than 102.000 individuals revealed that etoricoxib and diclofenac are the most potent treatment options for OA, and the etoricoxib is safer than diclofenac [da Costa et al.].

Authors’ answer:

Thank you for this this comment. After discussion, the authors agree with the reviewer and decided to rewrite some parts of this section.

R#1:

  1. The authors present in detail the benefits of PRP. However, other autologous biological therapies also act due to growth factors. Innovative and efficient treatment is the autologous micro-fragmented adipose tissue local injection. MFAT promoted the regeneration of cartilage, migration capacity of chondrocytes, and improvement of the subchondral bone lesions in animal models (Xu). Moreover, in human studies, MFAT improved the Tegner Lysholm Knee, VAS, IKDC-subjective, and the total KOOS scores (Russo). MFAT supplies mesenchymal stem cells, biological scaffolds, and growth factors to the joint. I suggest mentioning this therapeutical option, too.

Authors’ answer:

Thank you for this valuable comment. We added the information about MFAT in the knee OA therapy according to the reviewer’s suggestions.

R#1:

  1. There are in the manuscript some improper citations, like:

“A recent meta-analysis of randomized controlled trials (RCT) evaluating the effects of PPR or intra-articular hyaluronic acid (IA HA) injections in patients with knee OA documented that PRP is an effective treatment that reduced pain and improved functional status [35].” Please, correct them.

Authors’ answer:

Thank you for this valuable comment. Due to the suggestions we have corrected manuscript according to your recommendation.

R#1:

  1. Concerning the OARSI treatment guidelines, the newest version should be included among the references [Bannuru et. al.]

Authors’ answer:

To address your concerns, we have corrected appropriate references.

If these improvements are performed, I agree with the publication of this manuscript in IJMS.

Authors’ answer:

According to referee's suggestions we have reformulated appropriate parts. We would also like to thank you in a special way for indicating a list of valuable references. Appropriate corrections of the reference list were included in the new text. It will be great pleasure for our team, if You will find our manuscript adequate to publish in IJMS.

References

  1. Altman R, Hochberg M, Gibofsky A, Jaros M, Young C. Efficacy and safety of low-dose SoluMatrix meloxicam in the treatment of osteoarthritis pain: a 12-week, phase 3 study. Curr Med Res Opin. 2015 Dec;31(12):2331-43. doi: 10.1185/03007995.2015.1112772. PMID: 26503347.
  2. Bannuru RR, Osani MC, Vaysbrot EE, Arden NK, Bennell K, Bierma-Zeinstra SMA, Kraus VB, Lohmander LS, Abbott JH, Bhandari M, Blanco FJ, Espinosa R, Haugen IK, Lin J, Mandl LA, Moilanen E, Nakamura N, Snyder-Mackler L, Trojian T, Underwood M, McAlindon TE. OARSI guidelines for the non-surgical management of knee, hip, and polyarticular osteoarthritis. Osteoarthritis Cartilage. 2019 Nov;27(11):1578-1589. doi: 10.1016/j.joca.2019.06.011. Epub 2019 Jul 3. PMID: 31278997.
  3. Csifó (Vajda), E.; Nagy, E.E.; Horváth, E.; Fárr, A.-M.; Muntean, D.-L. Mid-term effects of meloxicam on collagen type II degradation in a rat osteoarthritis model induced by iodoacetate. Farmacia 2015, 63, 556–560
  4. da Costa BR, Pereira TV, Saadat P, Rudnicki M, Iskander SM, Bodmer NS, Bobos P, Gao L, Kiyomoto HD, Montezuma T, Almeida MO, Cheng PS, Hincapié CA, Hari R, Sutton AJ, Tugwell P, Hawker GA, Jüni P. Effectiveness and safety of non-steroidal anti-inflammatory drugs and opioid treatment for knee and hip osteoarthritis: network meta-analysis. BMJ. 2021 Oct 12;375:n2321. doi: 10.1136/bmj.n2321. PMID: 34642179; PMCID: PMC8506236.
  5. Nagy E, Vajda E, Vari C, Sipka S, Fárr AM, Horváth E. Meloxicam ameliorates the cartilage and subchondral bone deterioration in monoiodoacetate-induced rat osteoarthritis. PeerJ. 2017;5:e3185. Published 2017 Apr 12. doi:10.7717/peerj.3185
  6. Puljak L, Marin A, Vrdoljak D, Markotic F, Utrobicic A, Tugwell P. Celecoxib for osteoarthritis. Cochrane Database Syst Rev. 2017;5(5):CD009865. Published 2017 May 22. doi:10.1002/14651858.CD009865.pub2
  7. Russo A, Screpis D, Di Donato SL, Bonetti S, Piovan G, Zorzi C. Autologous micro-fragmented adipose tissue for the treatment of diffuse degenerative knee osteoarthritis: an update at 3 year follow-up. J Exp Orthop. 2018 Dec 19;5(1):52. doi: 10.1186/s40634-018-0169-x. PMID: 30569417; PMCID: PMC6300453.
  8. Wen ZH, Tang CC, Chang YC, Huang SY, Chen CH, Wu SC, Hsieh SP, Hsieh CS, Wang KY, Lin SY, Lee HL, Lee CH, Kuo HC, Chen WF, Jean YH. Intra-articular injection of the selective cyclooxygenase-2 inhibitor meloxicam (Mobic) reduces experimental osteoarthritis and nociception in rats. Osteoarthritis Cartilage. 2013 Dec;21(12):1976-86. doi: 10.1016/j.joca.2013.09.005. Epub 2013 Sep 29. PMID: 24084190.
  9. Xu T, Yu X, Yang Q, Liu X, Fang J, Dai X. Autologous Micro-Fragmented Adipose Tissue as Stem Cell-Based Natural Scaffold for Cartilage Defect Repair. Cell Transplant. 2019 Dec;28(12):1709-1720. doi: 10.1177/0963689719880527. Epub 2019 Sep 29. PMID: 31565996; PMCID: PMC6923561.

Reviewer 2 Report

Well done and thorough review on this topic. The work is well done and can be considered a summary of the current state of the art on the therapy of osteoarthritis.

TRANSLATE with x English
Arabic Hebrew Polish
Bulgarian Hindi Portuguese
Catalan Hmong Daw Romanian
Chinese Simplified Hungarian Russian
Chinese Traditional Indonesian Slovak
Czech Italian Slovenian
Danish Japanese Spanish
Dutch Klingon Swedish
English Korean Thai
Estonian Latvian Turkish
Finnish Lithuanian Ukrainian
French Malay Urdu
German Maltese Vietnamese
Greek Norwegian Welsh
Haitian Creole Persian  
TRANSLATE with COPY THE URL BELOW Back EMBED THE SNIPPET BELOW IN YOUR SITE Enable collaborative features and customize widget: Bing Webmaster Portal Back

Author Response

Reviewer #2: Comments to the Author

  1. Well done and thorough review on this topic. The work is well done and can be considered a summary of the current state of the art on the therapy of osteoarthritis.

Authors’ answer:

Thank you very much for this nice comment.

Reviewer 3 Report

This article is a review paper that systematically describes pharmacotherapy for osteoarthritis. At present, no fundamental treatment for osteoarthritis has been developed and it remains symptomatic. With limited symptomatic treatment options, the article provides details on NSAIDS, DMORD, and PRP for OA. There were no particular problems with the content, but there are a few minor points that need to be addressed.

In page 6, Figure 3, please correct the spelling of ''chodrotin'' to ''chondroitin''.

Page 9, the abbreviation for platelet-rich plasma is defined as PRP, but it is misspelled as PPR in several places in the text and needs to be corrected.

In the classification of cytokines on page 12, Table 2, the authors need to reconsider whether to include MMP9 and NF-kB, because MMP9 is a catabolite. This is because MMP9 is an enzyme and NF-kB is a transcription factor, not a cytokine.

Please check the list of abbreviations on page 15 below to make sure they are correct.

EGF: epidermal growth factor (Epithelial Growth factor)

NGF: Nerve growth factor (NFG)

NF-kB: Nuclear factor of kappa light chain enhancer of activated B cells

Author Response

Reviewer #3: Comments to the Author

This article is a review paper that systematically describes pharmacotherapy for osteoarthritis. At present, no fundamental treatment for osteoarthritis has been developed and it remains symptomatic. With limited symptomatic treatment options, the article provides details on NSAIDS, DMORD, and PRP for OA. There were no particular problems with the content, but there are a few minor points that need to be addressed.

  • In page 6, Figure 3, please correct the spelling of ''chodrotin'' to ''chondroitin''.
  • Page 9, the abbreviation for platelet-rich plasma is defined as PRP, but it is misspelled as PPR in several places in the text and needs to be corrected.

Authors’ answer:

Thank you for pointing out the mistake in this line, we have corrected it. Thank you for this valuable comment.

R#3:

  • In the classification of cytokines on page 12, Table 2, the authors need to reconsider whether to include MMP9 and NF-kB, because MMP9 is a catabolite. This is because MMP9 is an enzyme and NF-kB is a transcription factor, not a cytokine.

Authors’ answer:

Thank you for pointing out the mistake. The Table 2 was shortened in the new version of the manuscript.

R#3:

  • Please check the list of abbreviations on page 15 below to make sure they are correct.

EGF: epidermal growth factor (Epithelial Growth factor)

NGF: Nerve growth factor (NFG)

NF-kB: Nuclear factor of kappa light chain enhancer of activated B cells

Authors’ answer:

Thank you very much for this comment. Appropriate corrections were included in the new version of the manuscript.

Round 2

Reviewer 1 Report

The authors performed significant improvements in their manuscript, and their completions answer the issues raised in my comments.